# Decoupling the impact of microRNAs on translational repression versus RNA degradation in embryonic stem cells

Jacob W Freimer[1,2], TJ Hu[1,2], Robert Blelloch[1,2]*

[1]The Eli and Edythe Broad Center of Regeneration Medicine and Stem Cell Research, Center for Reproductive Sciences, University of California, San Francisco, San Francisco, United States; [2]Department of Urology, University of California, San Francisco, San Francisco, United States

**Abstract** Translation and mRNA degradation are intimately connected, yet the mechanisms that link them are not fully understood. Here, we studied these mechanisms in embryonic stem cells (ESCs). Transcripts showed a wide range of stabilities, which correlated with their relative translation levels and that did not change during early ESC differentiation. The protein DHH1 links translation to mRNA stability in yeast; however, loss of the mammalian homolog, DDX6, in ESCs did not disrupt the correlation across transcripts. Instead, the loss of DDX6 led to upregulated translation of microRNA targets, without concurrent changes in mRNA stability. The *Ddx6* knockout cells were phenotypically and molecularly similar to cells lacking all microRNAs (*Dgcr8* knockout ESCs). These data show that the loss of DDX6 can separate the two canonical functions of microRNAs: translational repression and transcript destabilization. Furthermore, these data uncover a central role for translational repression independent of transcript destabilization in defining the downstream consequences of microRNA loss.
DOI: https://doi.org/10.7554/eLife.38014.001

*For correspondence:
robert.blelloch@ucsf.edu

Competing interests: The authors declare that no competing interests exist.

## Introduction

Gene expression is determined through a combination of transcriptional and post-transcriptional regulation. While transcriptional regulation is well studied, less is known about how post-transcriptional events contribute to overall mRNA levels. Mammalian mRNAs display a wide range of half-lives ranging from minutes to over a day (*Schwanhäusser et al., 2011*). The wide range of mRNA stabilities are regulated by both intrinsic sequence features as well as the binding of regulatory factors such as microRNAs and RNA-binding proteins (*Cheng et al., 2017*; *Hasan et al., 2014*; *Wu and Brewer, 2012*). However, the identity and how such features and regulatory factors impact mRNA stability are not well understood.

One process that is intimately linked to mRNA stability is translation (*Roy and Jacobson, 2013*). Quality control mechanisms such as nonsense mediated decay, no go decay, and non-stop decay sense aberrant translation and lead to mRNA degradation (*Parker, 2012*; *Shoemaker and Green, 2012*). Translation initiation and elongation can also influence mRNA stability (*Huch and Nissan, 2014*). In yeast, inhibition of translation initiation through either 5' cap binding mutants or drug treatment leads to accelerated mRNA decay (*Chan and Mugler, 2017*; *Huch and Nissan, 2014*; *Schwartz and Parker, 1999*). Conversely, treatment with cycloheximide, which blocks ribosome elongation, stabilizes mRNAs (*Beelman and Parker, 1994*; *Chan and Mugler, 2017*; *Huch and Nissan, 2014*). The mechanisms linking translation to mRNA stability are poorly understood.

MiRNAs are small, non-coding RNAs that bind to the 3' UTR of their target transcripts to inhibit translation and/or induce mRNA destabilization (*Fabian and Sonenberg, 2012*; *Jonas and*

*Izaurralde, 2015*). Whether they primarily impact translation or mRNA degradation has been intensely debated (*Iwakawa and Tomari, 2015*; *Jonas and Izaurralde, 2015*). The impact of miR-NAs on the translation of endogenous transcripts has been measured using ribosome profiling, which measures the ratio of ribosome protected fragments to input mRNA (this ratio is termed translational efficiency). Simultaneous RNA-Seq and ribosome profiling experiments across a number of contexts show that miRNAs produce larger changes in mRNA levels than in translational efficiency, leading to the suggestion that mRNA destabilization is the dominant effect of miRNA repression (*Eichhorn et al., 2014*; *Guo et al., 2010*). However, in other studies, it has been suggested that miR-NAs primarily inhibit translation. For example, in the early zebrafish embryo, ribosome profiling and RNA-Seq show that miRNAs induce translational repression without mRNA destabilization (*Bazzini et al., 2012*). Furthermore, experiments using miRNA reporters to examine the kinetics of miRNA repression suggest that translational repression precedes mRNA destabilization (*Béthune et al., 2012*; *Djuranovic et al., 2012*). These studies raise the question of whether translational repression is the direct mode of miRNA-driven suppression with mRNA destabilization being a secondary consequence. To resolve this question, it is important to genetically separate the two functions. However, despite extensive research, it is not known whether it is possible to decouple miRNA-induced translational repression and mRNA destabilization of endogenous transcripts in a cell where both occur.

The RNA-binding protein DDX6 and its yeast homolog DHH1 are DEAD box helicases that localize to P-bodies and stress granules. These proteins have been implicated in both translational repression and mRNA destabilization, suggesting that they may link these two processes (*Coller and Parker, 2005*; *Presnyak and Coller, 2013*). Tethering experiments in yeast that lack DCP2 or DCP1 demonstrate that DHH1 can repress translation upstream and independent of enhancing decapping (*Carroll et al., 2011*; *Sweet et al., 2012*). Tethering experiments in human cells demonstrate that DDX6 also represses translation (*Kuzuoğlu-Öztürk et al., 2016*). Furthermore, DDX6 binds to components of the decapping complex, but exactly how this impacts translation and mRNA stability is unclear (*Ayache et al., 2015*; *Nissan et al., 2010*; *Tritschler et al., 2009*). Additionally, through interactions with the CCR4-NOT complex and the decapping complex, DDX6 is thought to be involved in miRNA-mediated translational repression, but its exact role is not fully understood (*Chen et al., 2014*; *Chu and Rana, 2006*; *Mathys et al., 2014*; *Rouya et al., 2014*).

Here, we sought to understand how mRNA stability changes are linked to translation changes during early mammalian development. It has been suggested that up to 70% of the molecular changes during mouse embryonic stem cell (ESC) differentiation are due to post-transcriptional regulation (*Lu et al., 2009*). Therefore, we measured and analyzed changes in mRNA stability and translational efficiency during ESC differentiation. Surprisingly, we found that the vast majority of molecular changes during this transition are driven by transcriptional, not post-transcriptional mechanisms. However, within self-renewing ESCs there was a wide range of mRNA stabilities. These stability differences correlated with translation levels. We generated *Ddx6* KO ESCs to determine whether DDX6 links translation to mRNA stability. Unlike its yeast homolog, DDX6 did not appear to play a general role in linking the two. However, its loss did lead to the translational upregulation of miRNA targets with little associated changes in mRNA stability. The resulting cells looked phenotypically and molecularly similar to cells deficient for all miRNAs. Therefore, the loss of DDX6 is able to separate the two central functions of miRNAs: translational repression and mRNA destabilization. Furthermore, these data show miRNA-induced translational repression alone can recapitulate many of the downstream consequences of miRNAs.

## Results

### Transcriptional changes drive expression changes during the ESC to EpiLC transition

Previous work suggested that up to 70% of the molecular changes that occur during early ESC differentiation are due to post-transcriptional events (*Lu et al., 2009*). In that work, differentiation was induced by expressing a shRNA to Nanog in ESCs grown in LIF. These conditions are associated with a heterogeneous population of cells (*Ivanova et al., 2006*). To revisit this question, we turned to a reporter system and an optimized differentiation protocol that enables the homogenous

differentiation of naive ESCs to formative epiblast like cells (EpiLC), which is representative of the transition from the pre- to post-implantation epiblast in vivo (*Chen et al., 2018*; *Krishnakumar et al., 2016*; *Parchem et al., 2014*) (*Figure 1A*). Using this system, we characterized the changes in mRNA expression, mRNA stability, and translation that occur during the transition. RNA-Seq showed 1890 genes significantly upregulated and 1532 genes significantly downregulated during the ESC to EpiLC transition (*Figure 1B and F*). Known naive markers were downregulated, while known primed markers were upregulated confirming robust differentiation (*Figure 1—figure supplement 1A*) (*Boroviak et al., 2015*).

To separate the contribution of transcription versus mRNA stability to changing mRNA levels during the ESC to EpiLC transition, we used metabolic labeling with 4-thiouridine (4sU) (*Dölken et al., 2008*; *Rabani et al., 2011*; *Windhager et al., 2012*). To measure transcription, nascent transcripts were labeled with a 30-min 4sU pulse, biotinylated, pulled down with streptavidin, and sequenced (4sU-Seq). To measure mRNA stability, total RNA-Seq was performed in parallel and the ratio of nascent RNA to total RNA was used to calculate relative stabilities for each gene (*Rabani et al., 2011*). To validate these findings, a subset of genes spanning a range of stabilities were measured using an alternative method where transcription was blocked with actinomycin D and mRNA levels followed over a time course by RT-qPCR (*Figure 1—figure supplement 1B and C*). The relative stabilities predicted by the two approaches were highly correlated. Given that the 4sU-Seq approach avoids the secondary effects associated with blocking all transcription, we used those data for genome-wide analysis (*Bensaude, 2011*; *Lugowski et al., 2018*). Surprisingly, the 4sU/total mRNA data showed very few changes in mRNA stability between the ESC and EpiLC states (*Figure 1C and F*). This lack of changes was not because of noise among the replicates, as biological replicates were well correlated (*Figure 1—figure supplement 1D*).

The general lack of changes in mRNA stability suggested that transcription is the dominant regulator of the changing mRNA levels during the ESC to EpiLC transition. Indeed, fold changes in total mRNA levels correlated extremely well with fold changes in 4sU-labeled nascent transcripts (Spearman's rho 0.88; $p<2.22*10^{-16}$) (*Figure 1D*). Thus, nascent transcription, not mRNA stability, underlies the mRNA changes associated with the ESC to EpiLC transition.

Next, we asked whether changes in translation play an important role in the ESC to EpiLC transition. To measure translation of all genes, we performed ribosome profiling to collect Ribosome Protected Footprints (RPFs) and matched total mRNA (*Ingolia et al., 2011*). As expected, RPFs showed a strong three nucleotide phasing of reads that was not present in the mRNA samples, confirming the quality of the data (*Figure 1—figure supplement 1E*). We calculated translation using the ratio of RPF/mRNA, also known as translational efficiency (*Ingolia et al., 2011*). Global analysis showed very few changes in translational efficiencies between the ESC and EpiLC states (*Figure 1E and F*). Biological replicates were well correlated showing that the overall lack of changes is not due to noise between the replicates (*Figure 1—figure supplement 1F*). Therefore, like mRNA stability, there are few changes in translational efficiency in early ESC differentiation.

## There is a wide range of RNA stabilities which are positively correlated with translation in ESCs

Although there were minimal changes in mRNA stability during the ESC to EpiLC transition, there was a wide range of mRNA stabilities within ESCs. For example, between the 25th and 75th percentile of mRNA stability, there was a 3.2-fold difference in stability and between the top and bottom 1% of mRNA stability there was over a 64-fold difference (*Figure 2A*). To identify features that explain the range of mRNA stabilities observed, we performed multiple linear regression taking into account the following features that previous studies implicated in affecting mRNA stability: 3′ UTR length, 5′ UTR length, CDS length, 3′ UTR GC content, 5′ UTR GC content, CDS GC content, AU-rich elements (ARE), miRNA-binding sites, number of exons in the transcript, and upstream ORFs (*Chan and Mugler, 2017*; *Cheng et al., 2017*; *Sharova et al., 2009*). Combined, these features explained 25% of the variation in mRNA stability. To identify which features had the greatest impact on stability, we analyzed the correlation between each individual feature and mRNA stability (*Figure 2B*). 3′ UTR length had the greatest impact and was negatively correlated with mRNA stability (Spearman's rho −0.3; $p<2.22*10^{-16}$) (*Figure 2C*). To validate the impact of 3′ UTRs on mRNA stability, we used a dual reporter system that contains a control GFP for normalization and a RFP with a cloned endogenous 3′ UTR from 12 representative genes (*Figure 2D*) (*Chaudhury et al.,*

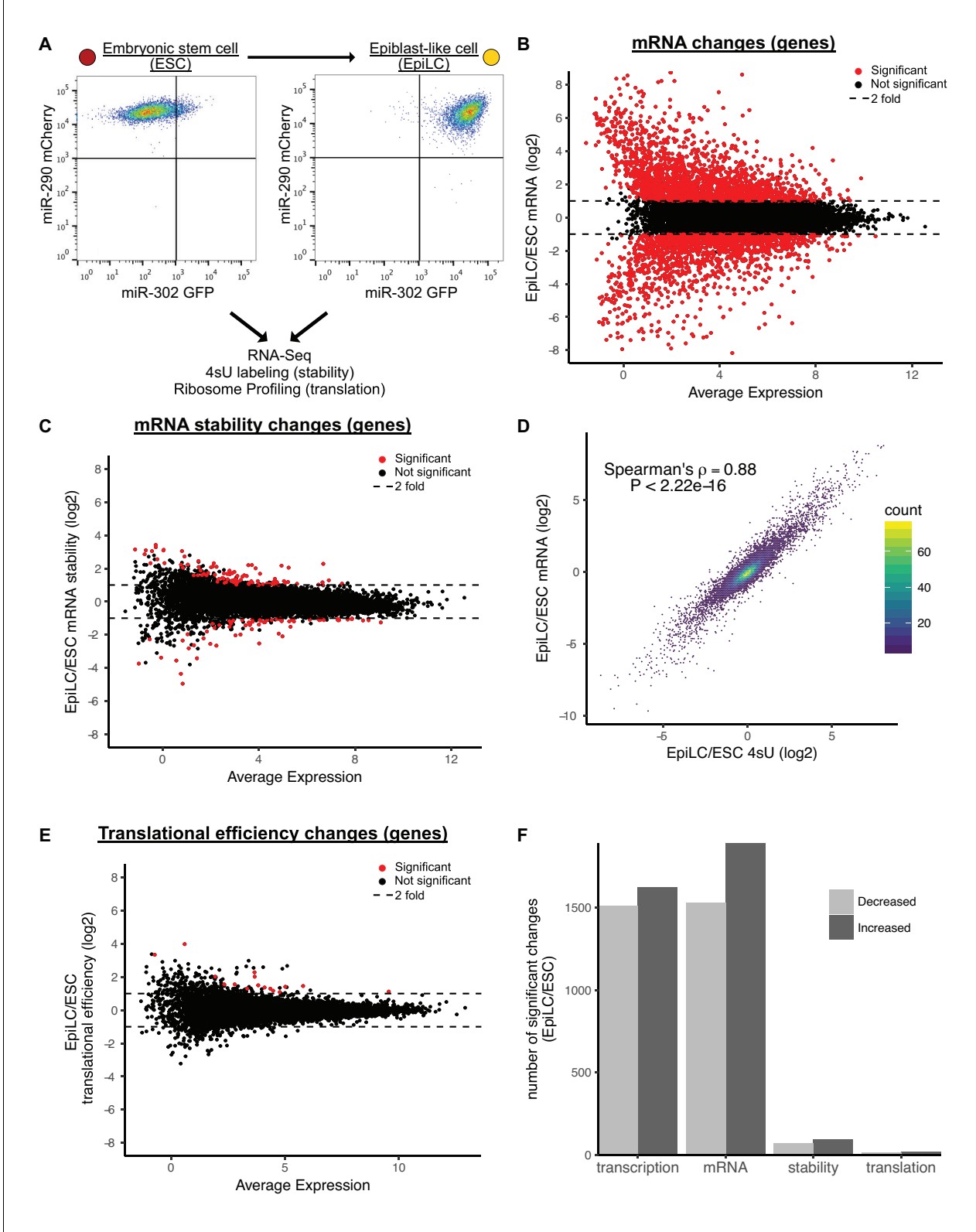

**Figure 1.** Transcriptional changes drive expression changes during the ESC to EpiLC transition. (A) Flow cytometry of the transition from naive embryonic stem cells (ESCs) (miR-302 GFP-, miR-290 mCherry+) to primed epiblast-like cells (EpiLCs) (miR-302 GFP+, miR-290 mCherry+). (B) MA plot of mRNA changes during the ESC to EpiLC transition. Significant changes are shown as red dots (Adjusted p value < 0.05 and |log2 fold change| > 1) in B, C, E. Dashed lines indicated a twofold change. (C) MA plot of mRNA stability changes during the ESC to EpiLC transition. (D) Correlation between

*Figure 1 continued on next page*

*Figure 1 continued*

changes in nascent transcription (4sU-labeled mRNA) and changes in mRNA levels during the ESC to EpiLC transition. The p value was calculated with correlation significance test. (E) MA plot of translational efficiency (TE) changes during the ESC to EpiLC transition. (F) The number of significant increases or decreases in transcription, mRNA levels, mRNA stability, and translational efficiency during the ESC to EpiLC transition. n = 3 for each ESC and EpiLC seq experiment. See also *Figure 1—figure supplement 1*.

DOI: https://doi.org/10.7554/eLife.38014.002

The following source data and figure supplements are available for figure 1:

**Figure supplement 1.** Validation of differentiation, mRNA stability measurements, and ribosome footprinting.

DOI: https://doi.org/10.7554/eLife.38014.003

**Figure supplement 1—source data 1.** Actinomycin D RT-qPCR data.

DOI: https://doi.org/10.7554/eLife.38014.004

*2014*). Flow cytometry analysis of cells expressing the reporter showed that the RFP/GFP ratio correlated well with the mRNA stability of the matching endogenous genes as measured by 4sU-Seq (*Figure 2D*).

MiRNAs are one regulatory factor that bind to the 3′ UTR of target mRNAs and recruit a complex of proteins that then destabilize the transcripts (*Fabian and Sonenberg, 2012*; *Jonas and Izaurralde, 2015*). In ESCs, the embryonic stem-cell-enriched cell cycle (ESCC) family of miRNAs represent a predominant fraction of all miRNAs in ESCs (*Greve et al., 2013*; *Houbaviy et al., 2003*; *Marson et al., 2008*). As expected, ESCC miRNA targets as a group were significantly less stable than all genes (p<2.22*10$^{-16}$, Mann-Whitney test) (*Figure 2—figure supplement 1A*). However, a large number of ESCC targets were still in the top 50% of the most stable genes (*Figure 2—figure supplement 1A*). These data suggest that, while miRNAs are strong destabilizers, they can only explain a small portion of the large range of mRNA stabilities seen in the cells.

Interestingly, analysis of the 4sU-Seq data showed that long non-coding RNAs (lncRNAs) were significantly less stable than protein coding genes (p<2.22*10$^{-16}$, Mann-Whitney test) (*Figure 2E*). This result suggested a strong role for translation in regulating RNA stability in ESCs. To further test this hypothesis, we performed polysome profiling (*Arava et al., 2003*). While both ribosome profiling and polysome profiling measure global levels of translation, polysome profiling can be a more sensitive measure of translational regulation (*Heyer and Moore, 2016*). We isolated monosome fractions, low polysome fractions containing 2–4 ribosomes, and high polysome fractions containing 4 + ribosomes and performed RNA-Seq (*Figure 2—figure supplement 1B*). To avoid confusion with the translational efficiency metric measured by ribosome profiling, we refer to the ratio of high polysome/monosome as translation levels. As expected, protein coding genes had a much higher translation level compared to lncRNAs (p<2.22*10$^{-16}$, Mann-Whitney test) (*Figure 2—figure supplement 1C*). We next compared the polysome profiling data to the mRNA stability data. There was a highly significant correlation between mRNA stability and translation levels across all genes (Spearman's rho 0.2; p<2.22*10$^{-16}$) (*Figure 2F*). We also observed a similar pattern when measuring translational efficiency with ribosome profiling data (*Figure 2—figure supplement 1D*). These findings suggested a direct link between translation level and mRNA stability.

Recent reports suggest that differential codon usage is a central mechanism in linking translation to mRNA stability (*Bazzini et al., 2016*; *Chan and Mugler, 2017*; *Cheng et al., 2017*; *Mishima and Tomari, 2016*; *Presnyak et al., 2015*). Therefore, we considered the possibility that codon optimality is a driving force in the wide range of mRNA stabilities. Codon optimality is driven in part through tRNA abundance, which is cell type specific in mammals and can alter translation and mRNA stability in a cell-type-specific manner (*Gingold et al., 2014*; *Goodarzi et al., 2016*). Unfortunately, tRNA abundance data does not exist for ESCs. Instead, we evaluated the frequency of codon usage between mRNAs with differing stabilities. This analysis uncovered small, but significant, differences in codon usage frequency between mRNAs in the top and bottom 20% of stabilities (*Figure 2—figure supplement 1E*). Therefore, codon optimality may in part explain the link between translation levels and mRNA stability.

## DDX6 regulates proliferation and morphology of ESCs

In yeast, the protein DHH1 has been shown to link translation to mRNA stability through codon optimality (*Radhakrishnan et al., 2016*). The mammalian homolog of DHH1, DDX6, has been shown to

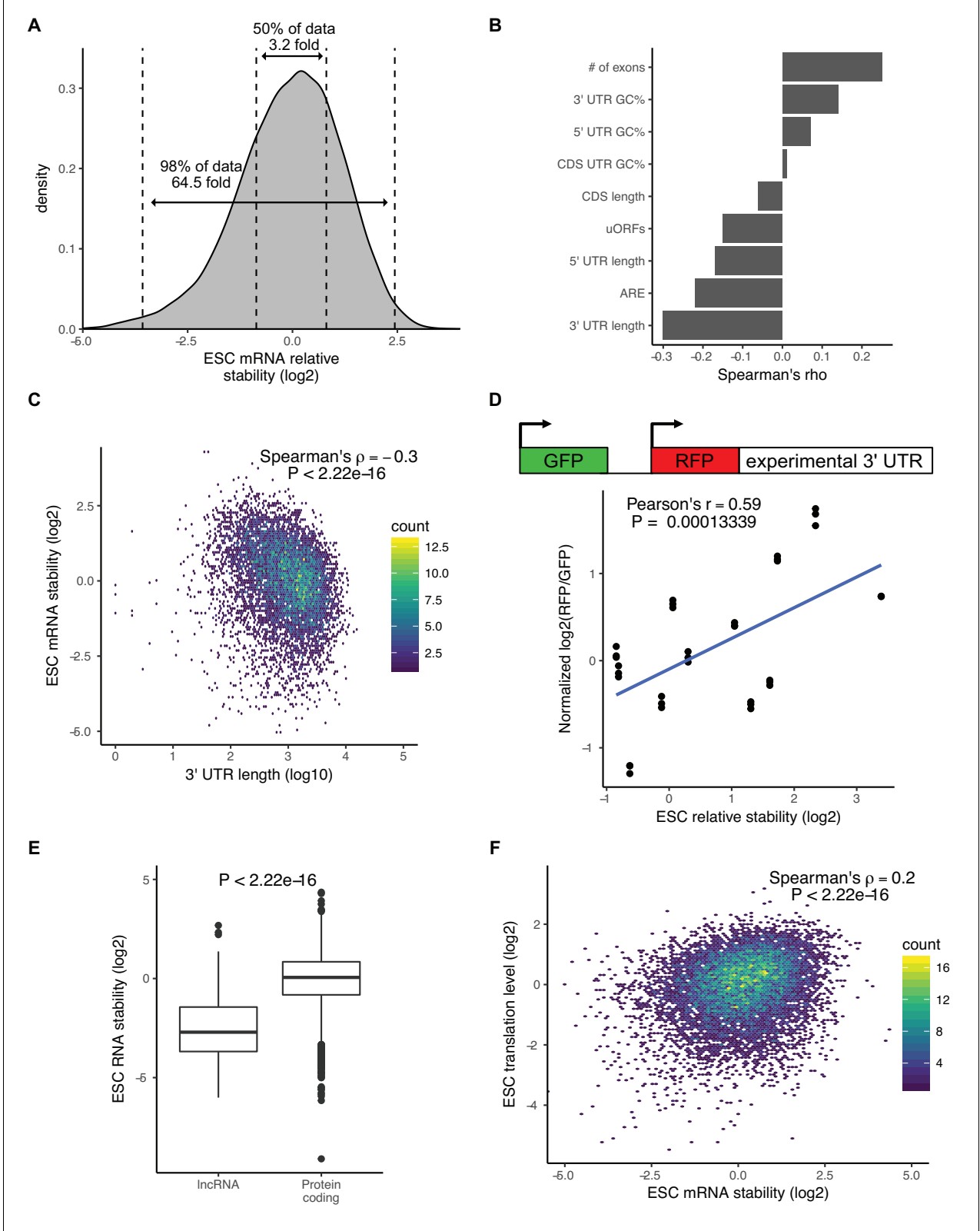

**Figure 2.** There is a wide range of RNA stabilities, which are positively correlated with translation level in ESCs. (A) The distribution of mRNA stabilities in ESCs. Dashed lines divide bottom 1%, middle 50%, and top 1% of the data. (B) The correlation between sequence features and mRNA stability in ESCs. uORFs (upstream open-reading frames), ARE (AU Rich Elements). (C) The correlation between 3' UTR length (log10) and mRNA stability in ESCs. (D) (Top) Schematic of dual reporter system to test endogenous 3' UTRs. (Bottom) Normalized median RFP/GFP ratios versus mRNA stability for

*Figure 2 continued on next page*

*Figure 2 continued*

endogenous genes as measured by 4sU-Seq. Clusters of dots indicate an endogenous 3' UTR, individual dots within a cluster represent biological replicates (n = 3). (E) RNA stability of long non-coding RNAs (lncRNAs) compared to protein-coding RNAs. The p value was calculated using the Mann–Whitney test. (F) Comparison between mRNA stability and translation level (high polysome/monosome ratio) in ESCs. The p value calculated with correlation significance test. n = 3. See also *Figure 2—figure supplement 1*.

DOI: https://doi.org/10.7554/eLife.38014.005

The following source data and figure supplement are available for figure 2:

**Source data 1.** RFP and GFP values for 3' UTR reporters.
DOI: https://doi.org/10.7554/eLife.38014.007
**Figure supplement 1.** Factors that affect RNA stability in ESCs.
DOI: https://doi.org/10.7554/eLife.38014.006

associate with both the mRNA decapping and deadenylation complex, also consistent with a potential link between mRNA stability and translation (*Chen et al., 2014*; *Mathys et al., 2014*; *Rouya et al., 2014*). Therefore, we next asked whether DDX6 may provide a mechanistic link for the relationship between translation and mRNA stability in ESCs. To investigate the function of DDX6 in ESCs, we produced *Ddx6* knockout (*Ddx6* KO) clones using CRISPR-Cas9. Sanger sequencing confirmed a single nucleotide insertion in one clone and a large deletion in a second clone, both of which produce a premature stop (*Figure 3—figure supplement 1A*). Western blot confirmed the absence of DDX6 protein in both clones (*Figure 3A*). We repeated the 4sU-Seq and polysome profiling in *Ddx6* KO and matched wild-type cells to measure changes in mRNA stability and translation levels. 4sU-Seq and total RNA-Seq showed that while there was a minimal reduction of nascent *Ddx6* mRNA, there was a drastic loss of mature *Ddx6* mRNA in the *Ddx6* KO cells (*Figure 3B*). This destabilization is consistent with nonsense-mediated decay and further validates the 4sU-Seq assay for assessing changes in mRNA stability.

The loss of DDX6 had little impact on the expression of pluripotency markers (*Figure 3C*). However, there were striking morphological changes in the cells (*Figure 3D*). Unlike wild-type ESCs which form tight domed colonies, *Ddx6* KO cells grew in a jagged, dispersed monolayer (*Figure 3D*). DDX6 localized to discrete punctate in the wild-type cells consistent with P-body localization, as previously reported (*Figure 3E*) (*Ernoult-Lange et al., 2012*; *Hubstenberger et al., 2017*; *Minshall et al., 2009*; *Presnyak and Coller, 2013*). Interestingly, the loss of DDX6 resulted in an abnormal distribution of the P-body marker DCP1a (*Figure 3F*). DDX6 loss also led to a reduction in proliferation in self-renewal culture conditions (*Figure 3G*). Together, these data show an important role for DDX6 in the formation and/or maintenance of P-bodies and in retaining normal cell morphology and proliferation.

## DDX6 separates miRNA-induced translational repression from RNA degradation

Since yeast DHH1 destabilizes lowly translated transcripts enriched in non-optimal codons, we expected that lowly translated genes might be stabilized in *Ddx6* KO ESCs (*Radhakrishnan et al., 2016*). However, there was minimal correlation between mRNA stability changes in *Ddx6* KO ESCs and wild-type translation levels (Spearman's rho $-0.11$; $p<2.22*10^{-16}$) (*Figure 4—figure supplement 1A*). The stabilized transcripts were not specifically enriched within the lowly translated transcripts and instead they occurred across all levels of translation. These data suggested that DDX6 does not link mRNA stability with translation levels across all genes. Next, we defined a set of codons as suboptimal based on their enrichment in unstable genes in wild-type ESCs and asked whether they are enriched among genes that are stabilized in *Ddx6* KO cells (*Figure 2—figure supplement 1E*). There was no enrichment (*Figure 4—figure supplement 1B*). Further comparing changes in median codon frequency in stable versus unstable transcripts in wild-type cells with changes in median codon frequency in stabilized versus unstabilized transcripts in *Ddx6* KO cells showed no correlation (*Figure 4—figure supplement 1C*). Species-specific tRNA adaptation index (stAI) provides an alternative metric of codon optimality. The stAI metric takes into account tRNA copy number and a tRNA's ability to wobble base pair with different codons (*Radhakrishnan et al., 2016*; *Sabi and Tuller, 2014*). We calculated stAI values for mouse and asked if they could predict changes in transcript stability associated with DDX6 loss. In contrast to the yeast homolog,

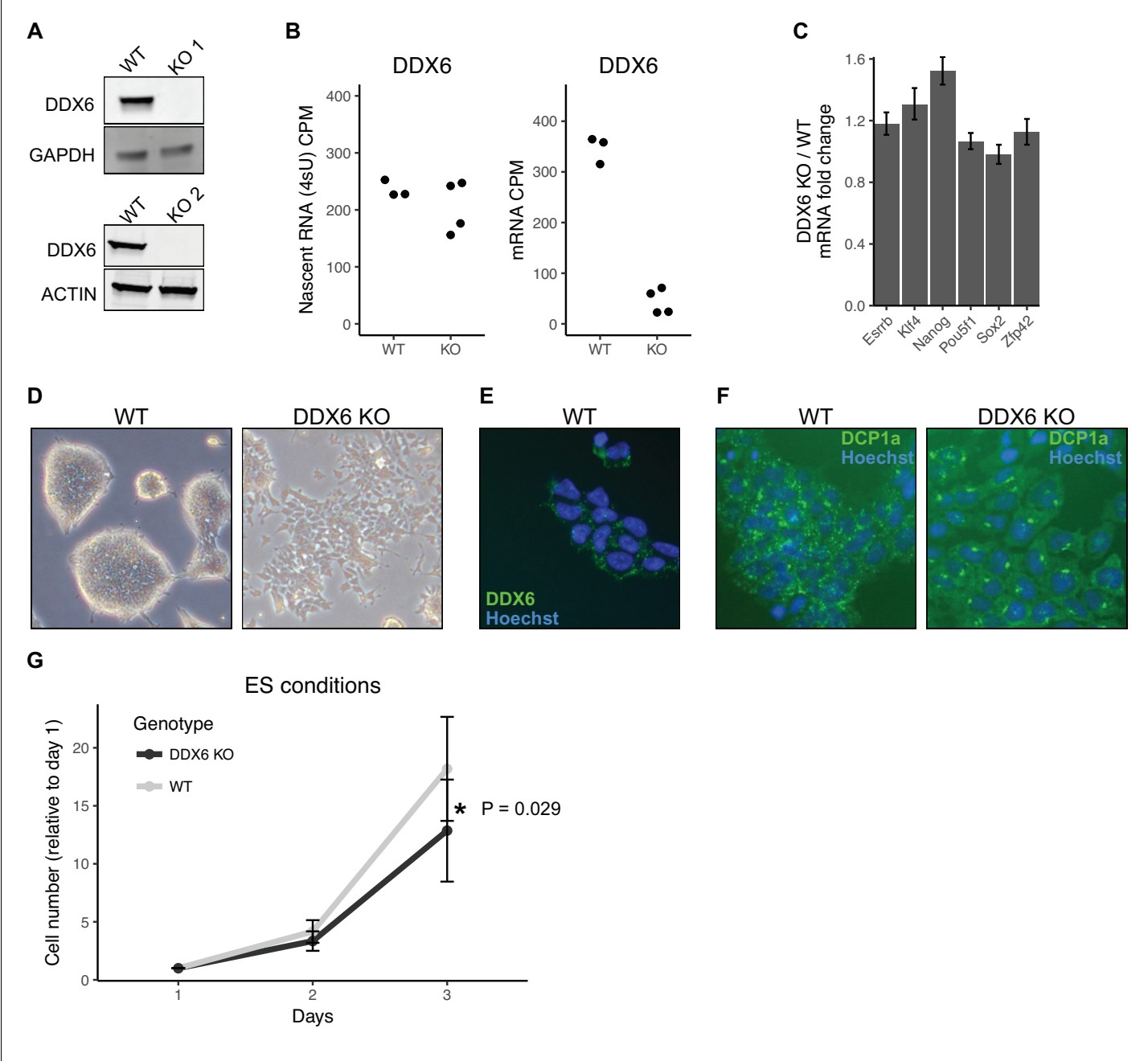

**Figure 3.** DDX6 regulates proliferation and morphology of ESCs. (A) Western blot of DDX6 in two *Ddx6* knockout (KO) lines. GAPDH and ACTIN were used as loading controls. (B) *Ddx6* counts per million (CPM) in nascent mRNA (4sU) or mRNA in wild-type (WT) and *Ddx6* KO cells. n = 3 for wild-type, n = 4 for *Ddx6* KO (2 replicates of each *Ddx6* KO line) (C) Expression of pluripotency genes in *Ddx6* KO ESCs based on RNA-Seq. Error bars represent 95% confidence interval. (D) Brightfield images of wild-type and *Ddx6* KO ESCs. Images taken at 20X. (E) DDX6 staining in wild-type ESCs. Images taken at 20X. (F) P-body staining against DCP1a in wild-type and *Ddx6* KO ESCs. Images taken at 20X. (G) Growth curves of wild-type and *Ddx6* KO ESCs in ESC maintenance conditions (LIF/2i). n = 6 for wild-type cells, n = 12 for *Ddx6* KO (six replicates of each *Ddx6* KO line). * indicates p<0.05 using a t-test, error bars are standard deviation. See also *Figure 3—figure supplement 1*.

DOI: https://doi.org/10.7554/eLife.38014.008

The following figure supplement is available for figure 3:

**Figure supplement 1.** Characterization of *Ddx6* KO ESCs.

DOI: https://doi.org/10.7554/eLife.38014.009

transcripts stabilized upon DDX6 loss did not correlate with low stAI values (*Figure 4—figure supplement 1D*). These data show that unlike yeast DHH1, the primary function of mammalian DDX6 is not to link codon optimality with transcript stability.

Several aspects of the *Ddx6* KO phenotype, including the cell morphology changes and growth defects, resemble the phenotype of *Dgcr8* KO cells (*Wang et al., 2007*). DGCR8 is essential for miRNA biogenesis and *Dgcr8* KO ESCs lack all miRNAs (*Wang et al., 2007*). DDX6 has been implicated as an effector of miRNA activity (*Chen et al., 2014*; *Chu and Rana, 2006*; *Mathys et al., 2014*; *Rouya et al., 2014*). Therefore, we next asked whether *Ddx6* KO cells have similar downstream molecular consequences as *Dgcr8* KO cells. To directly compare the two, we performed 4sU-Seq and polysome profiling in *Dgcr8* KO ESCs and analyzed the data in parallel with that of the *Ddx6* KO ESCs.

The ESCC family of miRNAs represent a predominant fraction of all miRNAs in ESCs (*Greve et al., 2013*; *Houbaviy et al., 2003*; *Marson et al., 2008*; *Melton et al., 2010*; *Wang et al., 2008*). They share the 'AAGUGC' seed sequence and thus have common downstream targets. Furthermore, re-introduction of a single member of the ESCC family of miRNAs can revert *Dgcr8* KO cells to a molecular phenotype highly similar to wild-type ESCs (*Gambardella et al., 2017*; *Melton et al., 2010*; *Wang et al., 2008*). Therefore, we chose to focus on the consequence of DGCR8 loss and DDX6 loss on these targets. As expected, the ESCC targets are stabilized relative to all genes in the *Dgcr8* KO cells (*Figure 4A*). However, these same targets showed little change in mRNA stability in the *Ddx6* KO cells (*Figure 4B*). Therefore, DDX6 does not appear to play a major role in transcript destabilization downstream of miRNAs.

The loss of DGCR8 also resulted in an increase in the translation levels of ESCC miRNA targets independent of its effect on stability, consistent with miRNAs both inhibiting translation and destabilizing transcripts (*Figure 4C*). In contrast to the stability data, the loss of DDX6 had a similar impact as the loss of DGCR8 on the translation levels of ESCC targets (*Figure 4D*). Indeed, *Dgcr8* KO and *Ddx6* KO affected the translation levels of individual targets to a similar extent (*Figure 4E*). These data show that DDX6 is an essential effector for miRNA-driven translational repression, but not mRNA destabilization. As such, DDX6 separates the two main functions of miRNAs showing that miRNA-driven translational repression and transcript destabilization are not dependent on one another.

## Translational repression alone underlies many of the downstream molecular changes associated with miRNA loss

Whether translational repression or mRNA destabilization is the predominant effect of miRNAs is controversial as it is difficult to separate the two (*Iwakawa and Tomari, 2015*; *Jonas and Izaurralde, 2015*). Given that the *Ddx6* KO cells retained mRNA destabilization, while losing translational repression of miRNA targets, we asked how well derepression of translation matches the downstream consequences of losing all miRNAs. Since the *Ddx6* KO and *Dgcr8* KO cells have partially overlapping phenotypes, we compared global changes in mRNA stability, mRNA levels, and translation levels. Strikingly, while there was little correlation in changes in mRNA stability, changes in both mRNA and translation levels were well correlated (*Figure 5*). Nascent transcriptional changes between *Ddx6* KO and *Dgcr8* KO measured by 4sU-Seq are also well correlated (*Figure 5—figure supplement 1A*) showing that the correlation in mRNA changes is due to transcriptional changes, likely secondary to the direct effects of *Ddx6* and *Dgcr8* loss on the translation of transcriptional regulators. These data show that translational repression alone can explain much of a miRNA's function in ESCs.

## Discussion

In this study, we sought to uncover how mRNA stability and translation are regulated within the ESC state and during differentiation. Using RNA-Seq, metabolic labeling (4sU-Seq), and ribosome profiling, we found that most changes during ESC differentiation are driven at the level of transcription. This finding contrasts Lemischka and colleagues' conclusion that post-transcriptional changes underlie many changes in protein levels during ESC differentiation (*Lu et al., 2009*). The difference may be explained by the different approaches used and the fact that Lemischka and colleagues focused their analysis on nuclear protein changes. Discordant changes between mRNA expression and

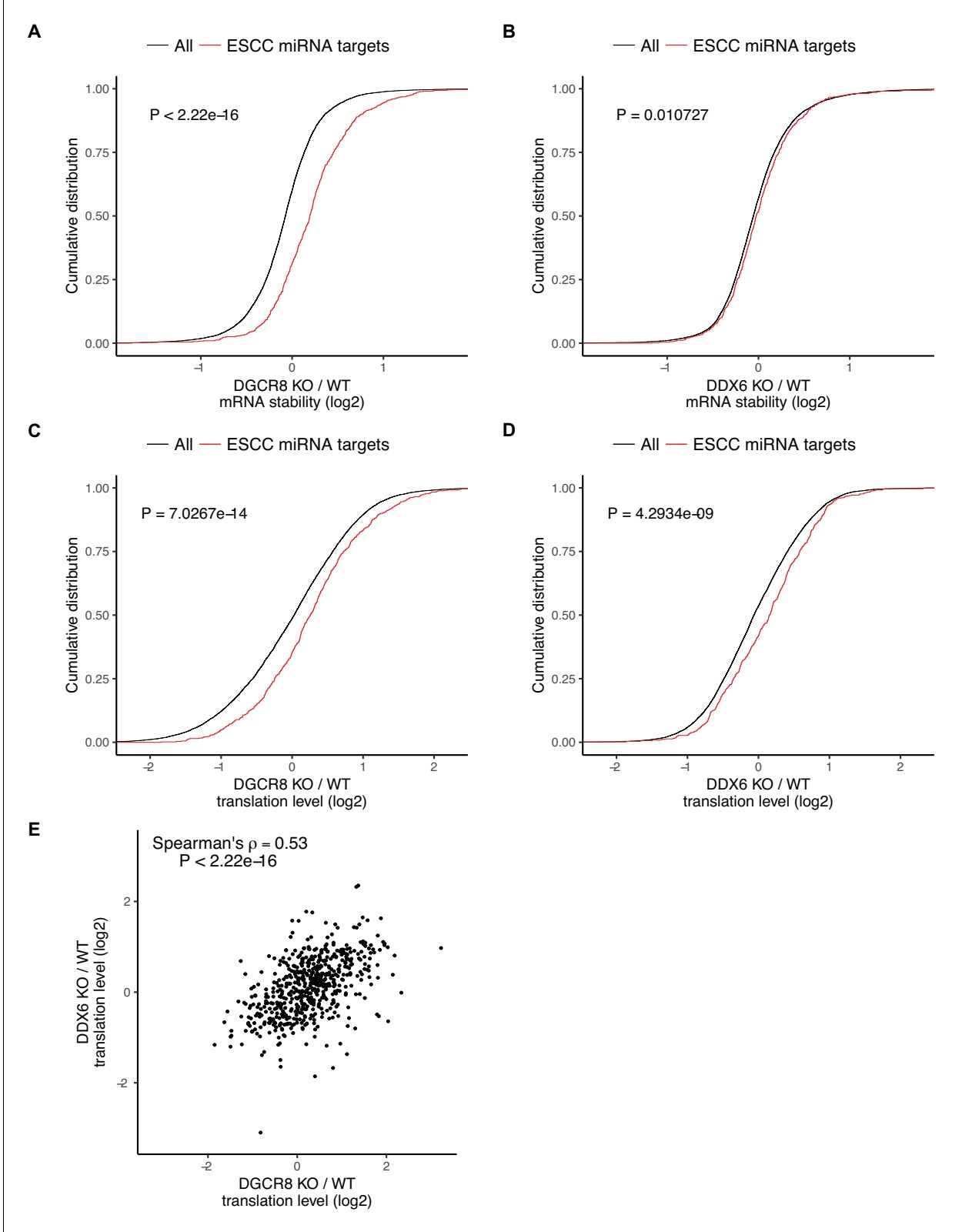

**Figure 4.** DDX6 separates miRNA-induced translational repression from RNA degradation. (**A–D**) mRNA stability or translation level changes of ESCC miRNA targets versus all mRNAs. The p value was calculated with Mann-Whitney test. A/B) mRNA stability changes in *Dgcr8* KO (**A**) or *Ddx6* KO (**B**) cells. n = 3 for wild-type, n = 4 for *Ddx6* KO (2 replicates of each *Ddx6* KO line), n = 3 for *Dgcr8* KO. C/D) Translation level changes in *Dgcr8* KO (**C**) or

*Figure 4 continued on next page*

*Figure 4 continued*

*Ddx6* KO (D) cells. n = 3 for each genotype. (E) Translation level changes of individual ESCC miRNA targets in *Dgcr8* KO and *Ddx6* KO ESCs. See also **Figure 4—figure supplement 1**.

DOI: https://doi.org/10.7554/eLife.38014.010

The following figure supplement is available for figure 4:

**Figure supplement 1.** Connection between stability changes and translation.

DOI: https://doi.org/10.7554/eLife.38014.011

nuclear protein levels could reflect changes in translational efficiency, protein stability, or protein localization (*Liu et al., 2016*). We measured translational efficiency, but not post-translational events; therefore, it remains plausible that protein degradation rates or protein localization are dynamic during ESC differentiation.

We found a positive correlation between mRNA stability and translation levels/efficiency in ESCs, similar to what other groups have observed recently in yeast (*Chan and Mugler, 2017*; *Heyer and Moore, 2016*; *Presnyak et al., 2015*). A number of mechanisms have been proposed. One is that ribosomes sterically hinder the degradation machinery from accessing the transcript. In support of this model, RNA-Seq of the 5' end of decapped RNA degradation intermediates shows a three nucleotide periodicity consistent with exonucleases running into the ribosome on a final round of translation (*Pelechano et al., 2015*). Alternatively, some RNA-binding proteins may sense slowly translating transcripts and accelerate their degradation as recently described for DHH1 (*Radhakrishnan et al., 2016*). However, in our data, the loss of the mammalian homolog of DHH1, DDX6, did not appear to link low levels of translation with low mRNA stability.

There has been extensive debate about whether miRNAs primarily inhibit translation or induce destabilization of their target transcripts (*Iwakawa and Tomari, 2015*; *Jonas and Izaurralde, 2015*). We measure translation as the ratio of polysome reads to monosome reads, which normalizes for any changes in mRNA levels. Therefore, we can independently quantify changes in translation level versus changes in mRNA stability. In *Dgcr8* KO ESCs, which lack all mature miRNAs, we observe that miRNAs both inhibit translation and induce mRNA destabilization of their targets within ESCs. In contrast, the loss of DDX6 only affects the translation of these miRNA targets (*Figure 5D*). That is, the number of protein molecules made per target transcript is increased, while the transcript stability remains the same. It has been suggested that translational repression of miRNA targets is the cause of mRNA destabilization or is at least a prerequisite (*Radhakrishnan and Green, 2016*). However, the *Ddx6* KO cells demonstrate that mRNA destabilization can occur independent of translational repression in a context where both forms of repression normally occur. Future studies will likely identify factors that can decouple translational repression and mRNA destabilization in the other direction so that miRNA targets are translationally repressed without inducing mRNA destabilization. This situation has been observed for synthetic miRNA reporters that cannot undergo deadenylation and subsequent degradation but can still be translationally repressed (*Kuzuoğlu-Öztürk et al., 2016*). MiRNA-induced translational repression in the absence of mRNA destabilization has also been observed in the early zebrafish embryo, but the mechanism underlying the phenomenon remains unclear (*Bazzini et al., 2012*)

Our data support a key role for DDX6 in the translational repression of endogenous miRNA targets. Several recent studies implicated DDX6 in the translational repression of miRNA reporters downstream of the CCR4-NOT complex (*Chen et al., 2014*; *Kuzuoğlu-Öztürk et al., 2016*; *Mathys et al., 2014*; *Rouya et al., 2014*). These studies found that the DDX6 RecA domain directly interacts with the CNOT1 MIF4G domain (*Chen et al., 2014*; *Mathys et al., 2014*; *Rouya et al., 2014*). Further, this interaction is important for the translational repression of both CNOT1 tethered reporters and miRNA reporters. In DDX6-depleted cells, repression of a miRNA reporter cannot be rescued by DDX6 mutants that cannot bind to CNOT1 (*Chen et al., 2014*; *Kuzuoğlu-Öztürk et al., 2016*; *Mathys et al., 2014*; *Rouya et al., 2014*). Conversely, CNOT1 requires binding to DDX6 in order to repress translation of a reporter that is resistant to deadenylation and degradation, suggesting that DDX6 can repress translation without affecting mRNA levels (*Kuzuoğlu-Öztürk et al., 2016*; *Mathys et al., 2014*). However, CNOT1 does not require DDX6 to degrade a reporter with a poly(A) tail, suggesting that CCR4-NOT can recruit degradation factors even when the interaction

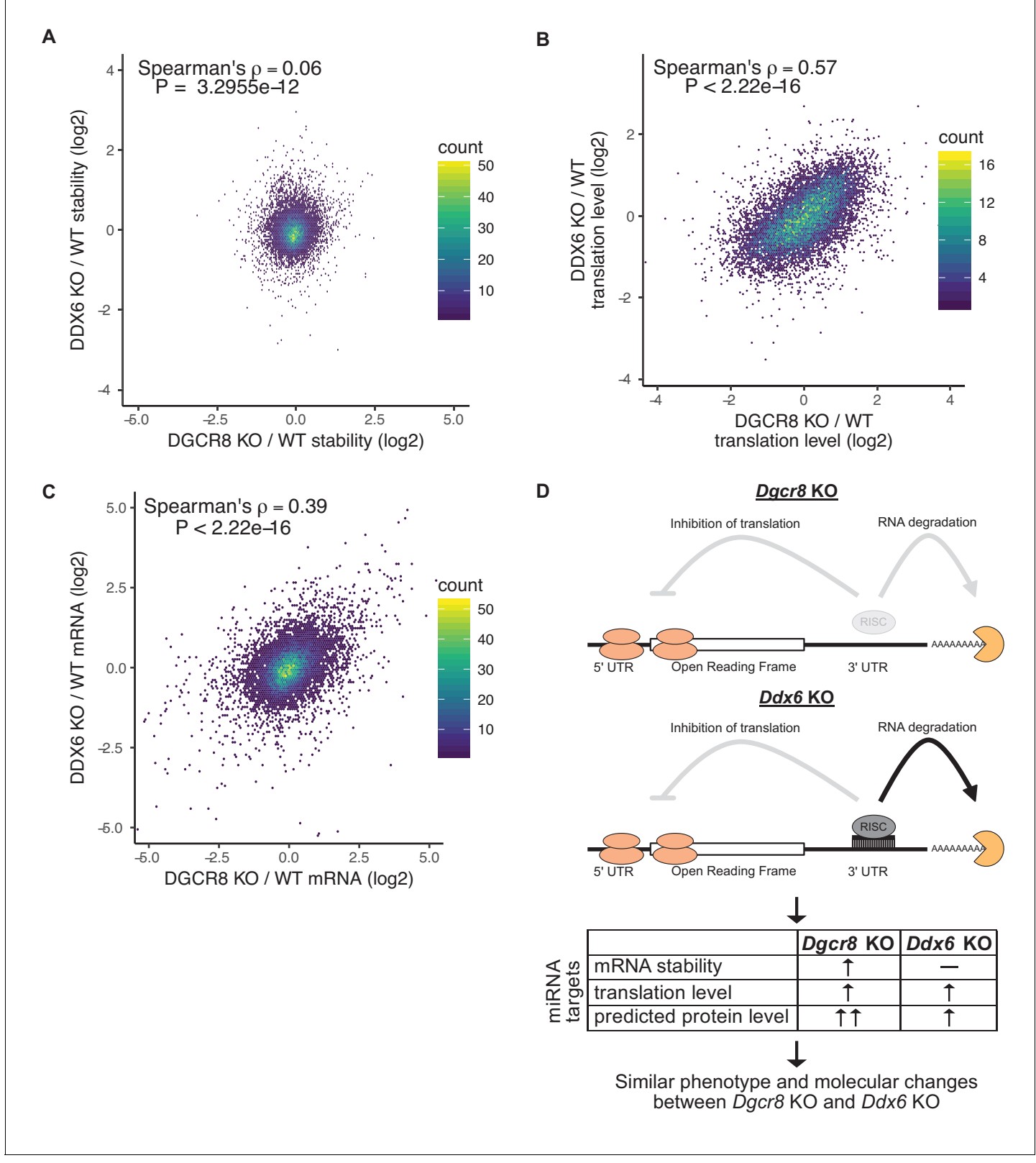

**Figure 5.** Translational repression alone underlies many of the downstream molecular changes associated with miRNA loss. (**A**) Comparison between mRNA stability changes in *Dgcr8* KO versus *Ddx6* KO cells. n = 3 for wild-type, n = 4 for *Ddx6* KO (2 replicates of each *Ddx6* KO line), n = 3 for *Dgcr8* KO. (**B**) Comparison between translation level changes in *Dgcr8* KO versus *Ddx6* KO cells. n = 3 for each genotype. (**C**) Comparison between mRNA changes in *Dgcr8* KO versus *Ddx6* KO cells. The p value was calculated with correlation significance test. (**D**) Summary schematic comparing *Dgcr8* KO

*Figure 5 continued on next page*

*Figure 5 continued*

cells to *Ddx6* KO cells. *Dgcr8* KO leads to the loss of both translational repression and mRNA destabilization of miRNA targets, while *Ddx6* KO only leads to the loss of translational repression of miRNA targets. mRNA stability is measured as the ratio of mRNA/4sU reads, changes in translation level are measured as the ratio of polysome/monosome reads, protein level changes are not directly measured but are predicted based on mRNA stability and translation level changes. Changes in translation level alone in *Ddx6* KO cells produce similar phenotypes and global molecular changes to *Dgcr8* KO cells. See also *Figure 5—figure supplement 1*.

DOI: https://doi.org/10.7554/eLife.38014.012

The following figure supplement is available for figure 5:

**Figure supplement 1.** Transcriptional changes in *Ddx6* KO and *Dgcr8* KO.

DOI: https://doi.org/10.7554/eLife.38014.013

with DDX6 is disrupted (*Kuzuoğlu-Öztürk et al., 2016*; *Mathys et al., 2014*). Direct tethering of DDX6 represses translation of a reporter and this repression is only mildly disrupted by mutations that prevent DDX6 from interacting with CNOT1 suggesting that DDX6 acts downstream of the CCR4-NOT complex (*Kuzuoğlu-Öztürk et al., 2016*). These studies also show that CNOT1 directly interacts with CNOT9, which binds to TNRC6, which in turn binds to AGO proteins (*Chen et al., 2014*; *Mathys et al., 2014*). This binding information directly links miRNA target recognition (AGO-mRNA binding) to translational repression through DDX6, via CNOT1-CNOT9-TNRC6 binding and is consistent with our findings.

It is not fully understood how DDX6 directly represses translation of miRNA targets or if it recruits additional effector molecules. The binding of CNOT1 changes the conformation of DDX6 and stimulates DDX6 ATPase activity (*Mathys et al., 2014*). This ATPase activity is essential for translational repression as DDX6 that contains mutations in the ATPase domain can no longer repress miRNA reporters (*Mathys et al., 2014*). However, how the ATPase domain contributes to translational repression is not known. Experiments using DDX6 tethered to different reporters suggest that DDX6 suppresses translational initiation independent of scanning (*Kuzuoğlu-Öztürk et al., 2016*). It was recently shown that DDX6 interacts with 4E-T, which competes with eIF4G for binding to the translation initiation factor eIF4E and leads to translational repression (*Kamenska et al., 2016*; *Ozgur et al., 2015*). Additionally, mutations in the FDF-binding domain of DDX6 prevents interaction with 4E-T and decapping proteins and prevents translational repression of a reporter (*Kuzuoğlu-Öztürk et al., 2016*). However, depletion of 4E-T only partially alleviates DDX6 mediated translational repression (*Kamenska et al., 2014*; *Kuzuoğlu-Öztürk et al., 2016*). Therefore, DDX6 likely interacts with additional unknown factors to inhibit translation initiation. Our data shows that the loss of DDX6 results in increased translation of miRNA targets to a similar level as the loss of all miRNAs, suggesting that DDX6 serves as a key link between the proteins that repress translation and the rest of the RNA-induced silencing complex.

Surprisingly, the loss of DGCR8 and DDX6 produce similar downstream consequences. Although not directly measured, protein levels of miRNA targets are likely higher in *Dgcr8* KO cells than in *Ddx6* KO cells as the former leads to both mRNA stabilization and translational derepression of miRNA targets, while the later only influences translation (*Figure 5D*). Yet, both knockout lines lead to similar morphology and proliferation defects as well as similar downstream molecular changes. It has been argued that mRNA changes are the dominant effect of miRNAs, since miRNA-induced changes in mRNA levels are often larger than changes in translational efficiency (*Eichhorn et al., 2014*; *Guo et al., 2010*). However, our data suggest that while miRNAs often have a significant effect on mRNA stability, their impact on translation alone can recapitulate a large portion of the downstream molecular and phenotypic effects associated with miRNA loss.

## Materials and methods

**Key resources table**

| Reagent type (species) or resource | Designation | Source or reference | Identifiers | Additional information |
|---|---|---|---|---|

*Continued on next page*

*Continued*

| Reagent type (species) or resource | Designation | Source or reference | Identifiers | Additional information |
|---|---|---|---|---|
| Gene (Mouse) | Ddx6 | NA | Ensembl: ENSMUSG00000032097 | |
| Gene (Mouse) | Dgcr8 | NA | Ensembl: ENSMUSG00000022718 | |
| Cell line (Mouse) | WT (V6.5) embryonic stem cell | PMID: 11331774; Novus Biologicals | NBP1-41162 | V6.5 mouse embryonic stem cell line from the Jaenisch lab, maintained in the Blelloch lab. Also available commercially from Novus Biologicals. |
| Cell line (Mouse) | Dgcr8 KO embryonic stem cell | PMID: 17259983; Novus Biologicals | NBA1-19349 | Dgcr8 KO mouse embryonic stem cell line previously generated in the Blelloch lab. Also available commercially from Novus Biologicals. |
| Cell line (Mouse) | Ddx6 KO embryonic stem cell | This paper | N/A | Ddx6 KO mouse embryonic stem cell line generated via CRISPR-Cas9 in the Blelloch lab from V6.5 parental cell line. |
| Cell line (Mouse) | miR290-mCherry and miR302-GFP reporter V6.5 embryonic stem cell | PMID: 26212322 | N/A | Mouse embryonic stem cell line used for differentiation in *Figure 1*. Previously generated in the Blelloch lab. |
| Antibody | anti-DDX6 | Bethyl Lab | A300-460A-T | 1:1000 |
| Antibody | anti-GAPDH | Santa Cruz Biotechnology | SC 25778 | 1:1000 |
| Antibody | anti-ACTIN | Sigma | A4700 | 1:1000 |
| Antibody | anti DCP1 | Abcam | ab47811 | 1:800 |
| Recombinant DNA reagent | pSpCas9(BB)—2A-GFP (PX458) plasmid | Addgene | 48138 | Used to generate DDX6 KO lines. |
| Recombinant DNA reagent | pBUTR(piggyBac-based 3' UnTranslated Region Reporter) plasmid | PMID: 24753411 | N/A | Used for 3' UTR reporter experiments. |
| Sequence-based reagent | CATGTGGTGATCGCTACCCC | This paper | N/A | DDX6 KO guide sequence |
| Commercial assay or kit | Ribo-Zero Gold kit | Illumina | MRZG126 | |
| Commercial assay or kit | KAPA Stranded RNA-Seq | KAPA | KK8400 | |
| Commercial assay or kit | KAPA HyperPrep Stranded RNA-Seq | KAPA | KK8540 | |
| Commercial assay or kit | Lexogen QuantSeq 3' FWD | Lexogen | 015.96 | |
| Commercial assay or kit | TruSeq Ribosome-Profiling | Illumina | RPHMR12126 | |
| Chemical compound, drug | 4-thiouridine (4sU) | Sigma | T4509-100MG | |
| Chemical compound, drug | Cycloheximide | Sigma | C4859-1ML | |
| Chemical compound, drug | Actinomycin D | Fisher Scientific | BP6065 | |
| Chemical compound, drug | MEK inhibitor PD0325901 | Peprotech | 3911091 | For naïve ESC culture |
| Chemical compound, drug | GSK3 inhibitor CHIR99021 | Peprotech | 2520691 | For naïve ESC culture |
| Other | Streptavidin Dynabeads | Invitrogen | 65305 | |

*Continued on next page*

*Continued*

| Reagent type (species) or resource | Designation | Source or reference | Identifiers | Additional information |
|---|---|---|---|---|
| Software, algorithm | Cutadapt version 1.14 | DOI: 10.14806/ ej.17.1.200 | RRID:SCR_011841 | |
| Software, algorithm | STAR version 2.5.3a | PMID: 23104886 | RRID:SCR_015899 | |
| Software, algorithm | Gencode M14 annotation | N/A | https://www.gencode genes.org | |
| Software, algorithm | Limma version 3.32.10 | PMID: 25605792 | RRID:SCR_010943 | |
| Software, algorithm | R version 3.4.2 | R Core Team | RRID:SCR_001905 | |
| Software, algorithm | ggplot2 version 2.2.1 | H. Wickham | RRID:SCR_014601 | |
| Software, algorithm | featureCounts version 1.5.3 | PMID: 24227677 | RRID:SCR_012919 | |

## Gene accession

The accession number for the sequencing data reported in this paper is GEO: GSE112767.

## 4sU-Sequencing

Samples were labeled with 500 uM 4-thiouridine (4sU) (Sigma) for 30 min then extracted with TRIzol (Invitrogen) and split into two groups. rRNA was depleted from Total RNA using the Ribo-Zero Gold kit (Illumina). 80 ug of RNA was biotinylated according to the following protocol *Rädle et al. (2013)*. Biotinylated 4sU RNA was isolated and washed using M-270 Streptavidin Dynabeads (Invitrogen), eluted with 100 mM DTT, and cleaned up with RNeasy minelute columns (Qiagen).

Libraries were generated with the KAPA Stranded RNA-Seq or Stranded HyperPrep library prep kit (Kapa) and sequenced with single-end 50 bp reads. Additional rounds of *Ddx6* KO and matched wild-type 4sU samples were sequenced with paired-end reads and counts were merged with single end reads.

## Cell culture and differentiation

ESCs were grown in Knockout DMEM (Invitrogen) supplemented with 15% Fetal Bovine Serum, LIF and 2i (Peprotech PD0325901 and CHIR99021). In order to generate EpiLCs, 400,000 ESCs were plated in a 15 cm plate; 24 hr later LIF/2i media was removed, cells were washed with PBS, and EpiLCs were collected ~56 hr later (*Krishnakumar et al., 2016*). Cells were tested to be free of mycoplasma.

## Quant Seq

QuantSeq 3' end counting was used for polysome profiling samples as well as matched wild-type, *Ddx6* KO, and *Dgcr8* KO mRNA samples (*Figure 5C*). RNA was isolated using RNeasy Micro kits (Qiagen). RNA-Seq libraries were generated using the QuantSeq 3' FWD kit (Lexogen) and sequenced with single-end 50 bp reads.

## Ribosome profiling

ESCs and EpiLCs were grown as above. Ribosome profiling libraries were generated using the Tru-Seq Ribosome Profiling kit (Illumina) and sequenced with single-end 50 bp reads. Three nucleotide periodicity of ribosome profiling reads was checked using RiboTaper (*Calviello et al., 2016*). Adapters were trimmed using cutadapt version 1.14 with the following settings: –minimum-length 26 –maximum-length 32 for the ribosome protected fragments or –minimum-length 32 for the total RNA. Adapter sequence used for trimming: AGATCGGAAGAGCACACGTCT. Reads were mapped with STAR version 2.5.3a to the mm10/Gencode M14 genome with the following settings: –outFilter-MultimapNmax 1 –outFilterMismatchNoverReadLmax 0.05 –seedSearchStartLmax 13 –winAnchor-MultimapNmax 200.

## Polysome profiling

Two plates of 6 million V6.5 ESCs were seeded in a 15 cm plate 48 hr prior to collection (*Eggan et al., 2001*). Cells were incubated with 100 ug/ml cycloheximide (Sigma) for 2 min and then moved to ice. Cells were washed and scraped in PBS with cycloheximide, spun down, and then lysed. Lysate was loaded onto a 10–50% sucrose gradient and centrifuged at 35,000 RPM for 3 hr. Gradients were collected on a gradient station (Biocomp). For each sample, the monosome, low polysome (2–4 ribosomes), and high polysome (4 + ribosomes were collected). RNA was extracted from gradient fractions with TRIzol LS (Invitrogen) and concentrated with the Zymo Clean and Concentrator-5 kit (Zymo) prior to library preparation with the QuantSeq 3' FWD kit (Lexogen).

## Western blot

Cells were collected in RIPA buffer with Protease Inhibitor Cocktail (Roche). Protein was run on a 4–15% gel (Bio-Rad) then transferred onto a PVDF membrane. Membranes were blocked with Odyssey blocking buffer, blotted with primary and secondary antibodies, and then imaged on the Odyssey imaging system (LI-COR). Antibodies: DDX6 1:1000 (A300-460A-T), GAPDH 1:1000 (SC 25778), ACTIN 1:1000 (A4700).

## Actinomycin D RT-qPCR

Cells were treated with 5 ug/ml Actinomycin D (Fisher Scientific). 0, 2, 4, 6, 8, and 12 hr after treatment, RNA was collected in TRIzol (Invitrogen). Reverse transcription was performed with the Maxima first strand synthesis kit (Thermo Scientific). qPCR was then performed with the SensiFAST SYBR Hi-ROX kit (Bioline) on an ABI 7900HT 384-well PCR machine. Each sample was normalized to 18S rRNA and its 0 hr time point.

## Cell count

50,000 cells were plated in multiple wells of a six well on day 0. On days 1, 2, and 3, cells were trypsinized and counted with a TC20 (Bio-rad). Day 2 and day 3 counts were normalized to the day 1 count.

## Imaging

Cells were fixed with 4% PFA 10 min at room temperature. Cells were blocked with 2% BSA and 1% goat serum in PBST. Cells were incubated with primary antibody for 1 hr at room temperature (Dcp1 abcam (ab47811) antibody 1:800 or DDX6 A300-460A) antibody 1:250). Cells were incubated with goat 488 secondary for 1 hr at room temperature. Cells were then imaged on a Leica inverted fluorescence microscope.

## Generation of *Ddx6* KO ESCs

*Ddx6* KO lines were generated using the protocol from (*Ran et al., 2013*). A guide RNA (CATGTGG TGATCGCTACCCC) was cloned into PX458, transfected into ESCs using Fugene 6, and then GFP-positive cells were sorted at clonal density. Clones were genotyped with the following primers (Fwd: CATTGCCCAGATTGAAGACA and Rvs: TCCTGACTGGCCTGAAACTT) and verified by western blot. Two different knockout clones were picked and used for all subsequent analysis.

## Species-specific tRNA adaptation index calculation

For each gene, the CDS region from the Gencode M14 annotation was used. Species-specific tRNA adaptation index (sTAI) values for each gene were calculated with stAIcalc (*Sabi et al., 2017*).

## Calculation of codon usage

For each gene, the APPRIS principle isoform was used to calculate codon usage frequency. To analyze differences in codon usage between stable and unstable genes, codon usage frequency was calculated for genes in the top 20% (stable) and bottom 20% (unstable) in terms of wild-type mRNA stability. For codon usage frequency for mRNA stability changes in *Ddx6* KO cells, we first filtered for genes in the bottom 20% of wild-type stability as defined above. Within those genes, we took the top 20% (top) and bottom 20% (bottom) of mRNA stability changes in *Ddx6* KO ESCs and calculated codon usage frequency within each group. Significant differences in codon frequency were

calculated using the Mann–Whitney test followed by Bonferroni correction. For the comparison between codon usage frequency in wild-type versus *Ddx6* KO, we took the median codon usage frequency in stable - the median codon usage frequency in unstable for each codon and compared it to the *Ddx6* KO median codon usage frequency in the bottom group - median codon usage frequency in the top group, using groups as defined above.

## Analysis software

For all samples, adapters were trimmed with Cutadapt version 1.14 (*Martin, 2011*) with the following options: -m 20 -a 'A{18}' -a 'T{18}' -a AGATCGGAAGAGCACACGTCTGAACTCCAGTCAC. Reads were mapped with STAR version 2.5.3a (*Dobin et al., 2013*) to the mm10/Gencode M14 genome with the following settings: –outFilterMultimapNmax 1 –outFilterMismatchNoverReadLmax 0.05 –seedSearchStartLmax 25 –winAnchorMultimapNmax 100. Reads were counted with featureCounts version 1.5.3 (*Liao et al., 2014*) using the Gencode M14 annotation with rRNA annotations removed with the following settings: -s. Differential expression was carried out with limma version 3.32.10 (*Ritchie et al., 2015*) and R version 3.4.2. Genes with a low number of reads were filtered out: a gene must have at least three counts per million across at least three replicates to be included for differential expression. For samples with multiple comparisons, a linear model was used for each condition in limma taking into account assay type (e.g. 4sU versus total RNA) and cell type (e.g. KO versus wild-type); significant changes in stability or translation are based on the interaction term. All downstream analyses were performed in R version 3.4.2 and plotted with ggplot2 version 2.2.1.

## Polysome profiling analysis

RNA-Seq from the monosome, low polysome (2–4 ribosomes), and high polysome (4 + ribosomes were collected) fractions was mapped as above. Translation level was defined as the ratio of the high polysome counts divided by the monosome counts. For KO versus wild-type analysis, a linear model was used for each condition in limma and significant changes in translation are based on the interaction term.

## 4sU-Seq analysis

By measuring transcription rate and steady state mRNA levels, it is possible to infer the relative degradation rate (*Rabani et al., 2011*). It is assumed that across the population of cells there is no change in mRNA levels over time for a given state. Therefore, changes mRNA levels can be modeled by their production rate $\alpha$ and degradation rate $\beta$.

$$dmRNA/dt = \alpha - \beta[mRNA] \approx 0$$

Solving for this equation, degradation rates can be calculated using a production rate (in this case nascent RNA transcription as measured by 4sU incorporation) and the concentration of total mRNA in the cell (as measured by total RNA-Seq).

$$\alpha/[mRNA] \approx \beta$$

For KO versus wild-type analysis, a linear model was used for each condition in limma and significant changes in stability are based on the interaction term.

## Analysis of features regulating RNA stability

For each gene with multiple isoforms, the APPRIS principle isoform was used. APPRIS data were downloaded on 10/30/2017. Log10(feature lengths), GC %, and log10(number of exons) were calculated in R version 3.4.2. Upstream open-reading frames were defined as the number of ATG sequences in the 5' UTR. AU-rich elements were defined as the number of UAUUUAU sequences in the 3' UTR. miRNA sites were defined as below. Each of these features and mRNA stability were used in a multiple linear regression using the lm function in R version 3.4.2. Additionally, the Spearman correlation was calculated between each feature and mRNA stability.

## microRNA targets

Conserved microRNA targets were downloaded from Targetscan mouse release 7.1. This list was filtered for genes that are targeted by the miR-291–3 p/294–3 p/295–3 p/302–3 p family yielding 765 target genes.

## 3' UTR analysis

For each gene, the APPRIS principle isoform was used to calculate log10 (3' UTR length). Log10(3' UTR length) was then compared to log2 relative mRNA stability.

## 3' UTR reporters

Endogenous 3' UTRs from the following genes were amplified from ESC cDNA: ENSMUSG00000021583, ENSMUSG00000029580, ENSMUSG00000043716, ENSMUSG00000010342, ENSMUSG00000021665, ENSMUSG00000024406, ENSMUSG00000052911, ENSMUSG00000058056, ENSMUSG00000020105, ENSMUSG00000026003, ENSMUSG00000020038, ENSMUSG00000025521, ENSMUSG00000031503. Genes were cloned into the pBUTR (piggyBac-based 3' UnTranslated Region reporter) using gateway cloning as outlined in *Chaudhury et al. (2014)*. Reporters were transfected into ESCs using Fugene 6 (Promega). Cells were treated with Geneticin to enrich for transfected cells. Cells were analyzed on an LSRII (BD). RFP+/GFP+ cells were gated in FlowJo and median RFP/GFP ratios were calculated. RFP/GFP ratios were standardized between days to accounts for differences in laser power.

## Primers

| Gene | Forward | Reverse |
| --- | --- | --- |
| Gapdh | AGGTCGGTGTGAACGGATTTG | TGTAGACCATGTAGTTGAGGTCA |
| Nanog | AACCAAAGGATGAAGTGCAAGCGG | TCCAAGTTGGGTTGGTCCAAGTCT |
| Zfp42 | CTCCTGCACACAGAAGAAAGC | CACTGATCCGCAAACACC |
| Fgf5 | CCTTGCGACCCAGGAGCTTA | CCGTCTGTGGTTTCTGTTGAGG |
| Otx2 | CAACTTGCCAGAATCCAGGG | GGCCTCACTTTGTTCTGACC |
| Bak | gctgacatgtttgctgatgg | gatcagctcgggcactttag |
| Ddx6 qPCR | ACTATACTCCGCTACTTTCCCTC | TGGCGCTCCGTTACATATG |
| 18S | GTGGAGCGATTTGTCTGGTT | CGCTGAGCCAGTCAGTGTAG |
| MycN | AGTGTGTCTGTTCCAGCTACTG | TTCATCTTCCTCCTCGTCATCC |
| Pgap1 | AGTACCCCGAGTACCAGAAAAT | TCGAACTTGCTTATAGCTTCCAG |
| Impact | GTGAAGAAATCGAAGCAATGGC | GGTACTCACTTGGCAACATCA |
| Cyr61 | AACGAGGACTGCAGCAAAAC | TTCTGACTGAGCTCTGCAGATC |
| Amotl2 | AGGGACAATGAGCGATTGCAG | CCTCACGCTTGGAAGAGGT |
| Ddx6 Genotyping | CATTGCCCAGATTGAAGACA | TCCTGACTGGCCTGAAACTT |

## Acknowledgements

We thank the following people for critical reading of the manuscript: Marco Conti, Stephen Floor, Raga Krishnakumar, Brian DeVeale, and Deniz Goekbuget. We also acknowledge Indiana University for access to their Mason cluster of computers, supported by the National Science Foundation (DBI #1458641). This project was funded by the National Institutes of Health (R01 GM101180, R01 GM122439) to RB, and a Genentech Predoctoral Research Fellowship to JWF.

## Additional information

### Funding

| Funder | Grant reference number | Author |
|---|---|---|
| National Institute of General Medical Sciences | GM101180 | Robert Blelloch |
| National Institute of General Medical Sciences | GM122439 | Robert Blelloch |

The funders had no role in study design, data collection and interpretation, or the decision to submit the work for publication.

### Author contributions

Jacob W Freimer, Conceptualization, Data curation, Formal analysis, Validation, Investigation, Visualization, Writing—original draft, Writing—review and editing; TJ Hu, Investigation; Robert Blelloch, Conceptualization, Resources, Supervision, Funding acquisition, Project administration, Writing—review and editing

### Author ORCIDs

Jacob W Freimer (iD) http://orcid.org/0000-0001-9239-2272
Robert Blelloch (iD) http://orcid.org/0000-0002-1975-0798

### Decision letter and Author response

Decision letter https://doi.org/10.7554/eLife.38014.018
Author response https://doi.org/10.7554/eLife.38014.019

## Additional files

### Supplementary files

• Transparent reporting form
DOI: https://doi.org/10.7554/eLife.38014.014

### Data availability

Sequencing data have been deposited in GEO under accession codes GSE112767.

The following dataset was generated:

| Author(s) | Year | Dataset title | Dataset URL | Database, license, and accessibility information |
|---|---|---|---|---|
| Freimer JW, Hu T, Blelloch R | 2018 | Decoupling the impact of microRNAs on translational repression versus RNA degradation in embryonic stem cells | www.ncbi.nlm.nih.gov/geo/query/acc.cgi?acc=GSE112767 | Publicly available at the NCBI Gene Expression Omnibus (accession no: GSE112767). |

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
