## [Decision Letter]

Thank you for submitting your article "Decoupling the impact of microRNAs on translational repression versus RNA degradation in embryonic stem cells" for consideration by *eLife*. Your article has been reviewed by James Manley as the Senior Editor, a Reviewing Editor, and three reviewers. The reviewers have opted to remain anonymous.

The reviewers have discussed the reviews with one another and the Reviewing Editor has drafted this decision to help you prepare a revised submission.

Summary:

This is a very interesting and well-documented study, which demonstrates that translational efficiency of mRNAs in ESCs is closely positively correlated with their stability. The authors further demonstrate that depletion of the RNA helicase DDX6 does not eliminate this correlation what contrasts with the situation in yeast. The most important finding of the work is the demonstration that knock-out of *Ddx6* in ESCs relieves translational repression induced by miRNAs and that the phenotype of *Ddx6* KO cells is similar to that of cells in which miRNA population is eliminated through inactivation of DGCR8. This data provides very strong support to the notion that translational repression by miRNAs may represent a decisive factor explaining their biological function (at least in ESCs), and that DDX6 is a major mediator of the translational repression by miRNAs. Additionally, the authors find that during ESC differentiation transcriptional changes drive gene expression changes, which contrasts with conclusions from previous studies performed in Lemischa's lab.

Essential revisions:

The paper would be strengthened by additional work and better explanation some of the data as follows:

1) The authors frequently note that lower mRNA stability is related to the lower translation efficiency. Could they show such relationship within their own datasets? Also, what would be the possible complications of the interpretation of their own TE data?

2) There is a confusing aspect in that the authors report upregulation in translation of miRNA targets without an increase in mRNA stability; this is mechanistically counterintuitive. If a miRNA-targeted mRNA is not destabilized, then a change in translational repression is anticipated, as has been reported in zebrafish embryos. In sharp contrast, the authors report that miRNA targets are still being destabilized but their translation rates are upregulated. How can an mRNA that is being destabilized have higher rates of translation? DDX6 contact with the CCR4-NOT complex has been linked to 'pure' translational repression; however, this is always assessed in the context of a miRNA-targeted reporter mRNA that cannot be deadenylated and is therefore highly stable. This is in contrast to endogenous miRNA targets that can be deadenylated and degraded. Overall, the conclusion that translational repression can be separated from mRNA destabilization needs to be better explained in reference to the existing literature.

3) The authors should include in the Discussion section a paragraph better describing previous work demonstrating the involvement of DDX6 in translational repression by miRNAs. Although some of the studies are mentioned in two sentences in the Introduction and Results section, it would also be interesting to discuss (for example, in the Discussion section), that the role of DDX6 is documented by structural studies (Chen et al., 2014, Mathys et al., 2014), importance of the CNOT1 ~ DDX6 interaction for activation of the DDX6 ATPase activity, which in turn is needed for translational repression (Mathys et al., 2014; Kuzyoglu-Ozturk et al., 2016). A better discussion of the DDX6 role would make the paper more interesting to the wide readership.

4) Subsection “There is a wide range of RNA stabilities which are positively correlated with translation in ESCs”. "To avoid confusion…", the authors use the term "translational levels" here to distinguish it from the translational efficiency metric measured by ribosome profiling. However, in subsection “Transcriptional changes drive expression changes during the ESC to EpiLC transition” and elsewhere they use the term "levels" for ribosome profiling data. Be consistent!

5) Subsection “DDX6 regulates proliferation and morphology of ESCs” and corresponding figures. It is difficult to appreciate how big are the differences in codon optimality between the top and the bottom 20% of mRNAs. Can some statistical significance be provided for this data? Or, give some information about which codons/tRNA are rare and which are not.

6) Subsection “Translational repression alone underlies many of the downstream molecular changes associated with miRNA loss”. It is concluded, "The correlation in mRNA changes is likely due to transcriptional secondary effects…." Any data supporting this? Evidence of an increase in pre-mRNA or IVS-containing reads?

7) Discussion section. A similar situation was also observed with reporters which are prevented to undergo decay (see, for example, Kuzuoglu-Ozturk et al., 2016).

8) Many references are listed multiple times in the references list.

---

## [Author Response]

Summary:This is a very interesting and well-documented study, which demonstrates that translational efficiency of mRNAs in ESCs is closely positively correlated with their stability. The authors further demonstrate that depletion of the RNA helicase DDX6 does not eliminate this correlation what contrasts with the situation in yeast. The most important finding of the work is the demonstration that knock-out of Ddx6 in ESCs relieves translational repression induced by miRNAs and that the phenotype of Ddx6 KO cells is similar to that of cells in which miRNA population is eliminated through inactivation of DGCR8. This data provides very strong support to the notion that translational repression by miRNAs may represent a decisive factor explaining their biological function (at least in ESCs), and that DDX6 is a major mediator of the translational repression by miRNAs. Additionally, the authors find that during ESC differentiation transcriptional changes drive gene expression changes, which contrasts with conclusions from previous studies performed in Lemischa's lab.

We thank the reviewers for their interest and helpful suggestions. We have incorporated their feedback and added new graphs as well as added new text, both of which have improved the manuscript. We discuss the specific changes below.

Essential revisions:The paper would be strengthened by additional work and better explanation some of the data as follows:1) The authors frequently note that lower mRNA stability is related to the lower translation efficiency. Could they show such relationship within their own datasets? Also, what would be the possible complications of the interpretation of their own TE data?

To complement Figure 2F showing the positive correlation between mRNA stability and translation level as measured by polysome profiling, we have made a similar scatter plot with translation efficiency as measured by the ribosome profiling data. Similar to when using translation level, translation efficiency shows a positive correlation with mRNA stability which is now shown in Figure 2—figure supplement 1D. We are not aware of complications of the interpretation of our translational efficiency data; if the reviewers think we are missing something we would be happy to take it into account.

2) There is a confusing aspect in that the authors report upregulation in translation of miRNA targets without an increase in mRNA stability; this is mechanistically counterintuitive. If a miRNA-targeted mRNA is not destabilized, then a change in translational repression is anticipated, as has been reported in zebrafish embryos. In sharp contrast, the authors report that miRNA targets are still being destabilized but their translation rates are upregulated. How can an mRNA that is being destabilized have higher rates of translation? DDX6 contact with the CCR4-NOT complex has been linked to 'pure' translational repression; however, this is always assessed in the context of a miRNA-targeted reporter mRNA that cannot be deadenylated and is therefore highly stable. This is in contrast to endogenous miRNA targets that can be deadenylated and degraded. Overall, the conclusion that translational repression can be separated from mRNA destabilization needs to be better explained in reference to the existing literature.

The translation rates measured are relative translation rates normalized for mRNA levels. We have now clarified this issue in the text.

3) The authors should include in the Discussion section a paragraph better describing previous work demonstrating the involvement of DDX6 in translational repression by miRNAs. Although some of the studies are mentioned in two sentences in the Introduction and Results section, it would also be interesting to discuss (for example, in the Discussion section), that the role of DDX6 is documented by structural studies (Chen et al., 2014, Mathys et al., 2014), importance of the CNOT1 ~ DDX6 interaction for activation of the DDX6 ATPase activity, which in turn is needed for translational repression (Mathys et al., 2014; Kuzyoglu-Ozturk et al., 2016). A better discussion of the DDX6 role would make the paper more interesting to the wide readership.

We thank the reviewers for these suggestions. We have revisited these papers and have expanded our discussion to better discuss the previous literature regarding the role of DDX6 in the translational repression of miRNA reporters and the interaction of DDX6 with the CCR4-NOT complex.

4) Subsection “There is a wide range of RNA stabilities which are positively correlated with translation in ESCs”. "To avoid confusion…", the authors use the term "translational levels" here to distinguish it from the translational efficiency metric measured by ribosome profiling. However, in subsection “Transcriptional changes drive expression changes during the ESC to EpiLC transition” and elsewhere they use the term "levels" for ribosome profiling data. Be consistent!

We apologize for this oversight. We have gone through the manuscript and changed the text in several places to only use translational efficiency when referring to ribosome profiling data and to only use translation levels when referring to polysome profiling data.

5) Subsection “DDX6 regulates proliferation and morphology of ESCs” and corresponding figures. It is difficult to appreciate how big are the differences in codon optimality between the top and the bottom 20% of mRNAs. Can some statistical significance be provided for this data? Or, give some information about which codons/tRNA are rare and which are not.

We have added statistical significance for differences in codon frequency using the Mann–Whitney test followed by Bonferroni correction.

Codon optimality is driven in part through tRNA abundance, which is cell type specific in mammals and can alter translation and mRNA stability in a cell type specific manner (Goodarzi et al., 2016; Gingold et al., 2014). Unfortunately, tRNA abundance data does not exist for ESCs making it impossible to definitively assign codons/tRNA as rare or not in ESCs. To get around this problem when analyzing the *Ddx6* KO data, we focused on codons that were enriched in unstable genes in wild-type cells and asked if they were enriched in genes that were stabilized in *Ddx6* KO cells. They were not (Figure 4—figure supplement 1B). Therefore, *Ddx6* does not appear to be destabilizing these transcripts secondary to their codon usage. We have clarified these points in the text.

6) Subsection “Translational repression alone underlies many of the downstream molecular changes associated with miRNA loss”. It is concluded, "The correlation in mRNA changes is likely due to transcriptional secondary effects…." Any data supporting this? Evidence of an increase in pre-mRNA or IVS-containing reads?

We have added a new figure, Figure 5—figure supplement 1A, showing that changes in nascent RNA (4sU-Seq) between *Dgcr8* KO and *Ddx6* KO are also well correlated. This data, together with the lack of correlation for stability changes (Figure 5A), support the claim that the correlated mRNA changes are due to transcriptional effects and not due to changes in post-transcriptional regulation. We have updated that sentence to include the new data and it now reads “Nascent transcriptional changes between *Ddx6* KO and *Dgcr8* KO measured by 4sU-Seq are also well correlated (Figure 5—figure supplement 1A) showing that the correlation in mRNA changes is due to transcriptional changes, likely secondary to the direct effects of *Ddx6* and *Dgcr8* loss on the translation of transcriptional regulators.”

7) Discussion section. A similar situation was also observed with reporters which are prevented to undergo decay (see, for example, Kuzuoglu-Ozturk et al., 2016).

We apologize for this oversight. We have added a sentence discussing these reporters and citing Kuzuoglu-Ozturk et al. This section now reads: “This situation has been observed for synthetic miRNA reporters that cannot undergo deadenylation and subsequent degradation but can still be translationally repressed (Kuzuoğlu-Öztürk et al., 2016). MiRNA induced translational repression in the absence of mRNA destabilization has also been observed in the early zebrafish embryo, but the mechanism underlying the phenomenon remains unclear (Bazzini et al., 2012).”

8) Many references are listed multiple times in the references list.

We apologize for this oversight. All duplicate references have been removed.